# Tensin Regulates Fundamental Biological Processes by Interacting with Integrins of Tonsil-Derived Mesenchymal Stem Cells

**DOI:** 10.3390/cells11152333

**Published:** 2022-07-29

**Authors:** Gi Cheol Park, Ji Min Kim, Sung-Chan Shin, Yong-il Cheon, Eui-Suk Sung, Minhyung Lee, Jin-Choon Lee, Byung-Joo Lee

**Affiliations:** 1Department of Otolaryngology–Head and Neck Surgery, Samsung Changwon Hospital, Sungkyunkwan University School of Medicine, Changwon 51353, Korea; uuhent@skku.edu; 2Department of Otorhinolaryngology–Head and Neck Surgery, College of Medicine, Pusan National University and Biomedical Research Institute, Pusan National University Hospital, Busan 49241, Korea; jimin-kim@pusan.ac.kr (J.M.K.); shinsc0810@pusan.ac.kr (S.-C.S.); skydragonone@pusan.ac.kr (Y.-i.C.); 3Department of Otorhinolaryngology–Head and Neck Surgery, College of Medicine, Pusan National University and Biomedical Research Institute, Pusan National University Yangsan Hospital, Yangsan 50612, Korea; sunges@pusan.ac.kr (E.-S.S.); weichwein@pnuyh.co.kr (M.L.); ljc020971@pusan.ac.kr (J.-C.L.)

**Keywords:** tensin, proliferation, differentiation, focal adhesion, tonsil-derived mesenchymal stem cell

## Abstract

Human tonsil-derived mesenchymal stem cells (TMSCs) have a superior proliferation rate and differentiation potential compared to adipose-tissue-derived MSCs (AMSCs) or bone-marrow-derived MSCs (BMSCs). TMSCs exhibit a significantly higher expression of the tensin3 gene (*TNS3*) than AMSCs or BMSCs. TNS is involved in cell adhesion and migration by binding to integrin beta-1 (ITG β1) in focal adhesion. Here, we investigated the roles of four TNS isoforms, including TNS3 and their relationship with integrin in various biological processes of TMSCs. Suppressing TNS1 and TNS3 significantly decreased the cell count. The knockdown of TNS1 and TNS3 increased the gene and protein expression levels of p16, p19, and p21. TNS1 and TNS3 also have a significant effect on cell migration. Transfecting with siRNA TNS3 significantly reduced Oct4, Nanog, and Sox-2 levels. Conversely, when TNS4 was silenced, Oct4 and Sox-2 levels significantly increase. TNS1 and TNS3 promote osteogenic and adipogenic differentiation, whereas TNS4 inhibits adipogenic differentiation of TMSCs. TNS3 is involved in the control of focal adhesions by regulating integrin. Thus, TNS enables TMSCs to possess a higher proliferative capacity and differentiation potential than other MSCs. Notably, TNS3 plays a vital role in TMSC biology by regulating ITGβ1 activity.

## 1. Introduction

Focal adhesions are multi-protein assemblies in which transmembrane communication occurs at cell–matrix junctions. They function as a mechanical link that connects actin in the cell to the extracellular substrate and a channel to transmit biochemical signals between the cell and extracellular matrix (ECM). Therefore, focal adhesion plays a vital role in maintaining cell homeostasis in response to external stimuli or internal changes in cells [1,2].

The central structure of focal adhesion, integrins, are heterodimers composed of α and β subunits and transmembrane glycoproteins located at focal adhesions. The extracellular domain of integrin binds to substrates, such as fibronectin or proteoglycan, and the cytoplasmic domain is linked to actin filaments. Integrins structurally attach cells to the ECM [3]. Furthermore, the structures of the α and β subunits change after chemical or mechanical stimulation from inside or outside the cell, resulting in the transmission of biosignals in and out of the cell [4]. By performing these two basic functions, integrin plays an essential role in various biological activities of the cell, including cell migration, proliferation, and division [5,6,7]. Unlike growth factor receptors, integrin does not have intrinsic catalytic activity due to its lack of a tyrosine kinase domain in its structure. Therefore, it operates through collaboration with an extensive collection of cytoplasmic proteins called focal adhesion molecules [1,2]. Focal adhesion molecules are highly dynamic structures that attach to and detach from integrin, connect actin and integrin, and transmit signals from integrin into the cell. To date, the roles of various focal adhesion molecules, including vinculin, paxillin, talin, α-actinin, zyxin, vasodilator-stimulated phosphoprotein (VASP), focal adhesion kinase (FAK), and p130Cas, and their association with integrin have been gradually elucidated. However, further research is required to elucidate their complicated mechanisms [8,9].

Proliferative and differentiation potential are critical factors in the clinical application of stem cells. Human palatine tonsil-derived mesenchymal stem cells (TMSCs) have a higher proliferation rate and differentiation potential than those of bone-marrow-derived MSCs (BMSCs) or adipose-tissue-derived MSCs (AMSC) [10,11,12]. In our previous study, we reported that the expression of tensin3 (*TNS3*) was significantly higher in TMSCs than in BMSCs or AMSCs and that TNS3 influenced the high proliferative capacity and differentiation potential of TMSC by regulating integrin beta-1 (ITG β1) activity [13].

The TNS family comprises four isoforms of focal adhesion proteins involved in cell adhesion and migration by competitively binding to the cytoplasmic tail of ITG β1 against other integrin adaptor proteins [14]. However, studies on TNSs are limited compared to those on other focal adhesion molecules such as talin and kindlin. We revealed the role of TNS3 in focal adhesion and its relationship with integrin in a previous study [13]. However, the roles of the remaining three TNS isoforms have not been investigated. In this study, we comprehensively investigated the role of the TNS family of proteins in the focal adhesion of TMSCs.

## 2. Materials and Methods

### 2.1. Generation of Palatine TMSCs

The tonsils of four patients with chronic tonsillitis were collected during tonsillectomy after obtaining informed consent. Immediately after surgery, the tonsils were washed several times with phosphate-buffered saline (PBS). The tissue was digested in 0.075% collagenase type I (Sigma-Aldrich, St. Louis, MO, USA) at 37 °C to isolate the stem cells. Enzyme activity was inhibited by incubating the cells in alpha-modified Eagle’s medium (α-MEM) containing 10% fetal bovine serum (FBS) for 30 min, and the sample was centrifuged at 1200× *g* for 10 min. The pellet was filtered through a 100 μm nylon mesh to remove cell debris, and cell adhesion was confirmed by incubation for 1–2 days at 37 °C in a 5% CO_2_ atmosphere in α-MEM-containing 10% FBS, 100 U/mL penicillin, and 100 µg/mL streptomycin. The cells adhered stably to the plate, and the plate was washed with PBS to remove residual non-adherent cells. The resulting cell population was further maintained. All study protocols were reviewed and approved by the Institutional Review Board.

### 2.2. Quantitative Reverse Transcription Polymerase Chain Reaction and RNA Interference

mRNA expression was confirmed using quantitative PCR. Primers for *TNS1*, *TNS2*, *TNS3*, and *TNS4* were obtained from Bio-Rad (Hercules, CA, USA). The primer sequences for p16, p19, p21, Cyclin E, *CDC25*, Oct-4, *Nanog*, Sox-2, osteocalcin, *ALP* (alkaline phosphatase), peroxisome proliferator-activated receptor-gamma (*PPARγ*), and lipoprotein lipase (*LPL*) are shown in Table 1. All primer sequences were selected according to the established PubMed GenBank sequence. *GAPDH* was used as an internal control. Small interfering RNA (siRNA) oligonucleotide duplexes targeting *TNS1*, *2*, *3*, and *4* mRNA were purchased from ON-TARGETplus SMARTpool (Dharmacon, Thermo Fisher Scientific, Waltham, MA, USA; a nontargeted oligonucleotide duplex was also used). Transfection was performed according to the instructions of the Dharma FECT Transfection Reagent. Real-time quantification was conducted on LightCycler (Roche, Basel, Switzerland) using a fluorescent SYBR Green I PCR mixture. PCR kinetics and quantitative data were analyzed using LightCycler version 3.3 software.

### 2.3. Western Blotting

Protein concentration was determined using a BCA protein assay kit (Thermo Fisher Scientific). Total protein equivalents for each sample were separated by sodium dodecyl sulfate-polyacrylamide gel electrophoresis using 10% acrylamide gels; the proteins were transferred to polyvinylidene fluoride membranes. The membrane was immediately placed into a blocking buffer containing 5% bovine serum albumin for 1 h. The membrane was incubated with anti-p16, 19, 21, SOX-2, talin1, and kindlin 1 antibodies (1:2000; Santa Cruz Biotechnology, Dallas, TX, USA and Cell Signaling Technology, Danvers, MA, USA) at 4 °C overnight. Anti-GAPDH antibody (1:2000; Santa Cruz Biotechnology) was used as an internal control. After three 10 min washes, the membranes were incubated with anti-mouse and anti-rabbit peroxidase-conjugated secondary antibodies (1:10,000 dilution) for 1 h at 21 °C. Antibody labeling was detected using West-zol Plus and chemiluminescence Fluorchem^TM^SP (Alpha Innotech Corporation, San Leandro, CA, USA).

### 2.4. Cell Proliferation

Cell proliferation was measured using the cell counting kit-8 (CCK8) assay. After culturing the cells in a medium containing CCK8 solution (Dojindo Laboratories, Kumamoto, Japan) at 37 °C for 1 h, absorbance at 450 nm was measured with a spectrophotometer. This experiment was repeated four times.

### 2.5. Cell-Cycle Analysis

For cell-cycle analysis, 6-well plates were seeded with 5 × 10^4^ cells/well and treated with recombinant human fibroblast growth factor 5 (FGF5) at 37 °C for 48 h. The cells were harvested using 0.05% trypsin solution and centrifuged at 10,000× *g* for 15 min. The pellet was washed twice with Hank’s Balanced Salt Solution buffer and fixed in 70% ethanol overnight at −20 °C. The next day, the ethanol was removed, and the cells were resuspended in 500 mL of PBS containing 1 mg/mL propidium iodide and 100 µg RNase/mL; after incubation for 20 min, flow cytometry analysis was performed using a FACS Calibur (Becton Dickinson, San Jose, CA, USA).

### 2.6. Cell Migration Assay

Cell migration was analyzed using a transwell chamber with a pore size of 8 μm. Cells were seeded into the upper chamber at 4 × 10^5^/well, and IFN’- and TNF’-treated medium was dispensed into the lower chamber. The cells were incubated at 37 °C for 24 h to allow migration between the transwells by inflammatory cytokines. Cells that migrated and adhered to the lower chamber were washed, fixed with 10% formaldehyde, and stained with DAPI to measure cell migration. Stained cells were counted using a fluorescence microscope (Leica Microsystems, Wetzlar, Germany).

### 2.7. Adipogenic and Osteogenic Differentiation

To confirm that the TMSC cells separated from the tonic chamber were stem cells, we examined their ability to differentiate into bone formation and adipogenesis systems. TMSCs were cultured in medium containing lipogenesis-induced reagents (10% FBS, and 1 μM dexamethasone (Sigma-Aldrich, St. Louis, MO, USA) α-MEM, 100 μg/mL 3-isobutyl-L-methylxanthine, 5 μg/mL insulin, and 60 μM) to induce lipocyte differentiation. Oil Red O (Sigma-Aldrich) staining was performed after treatment with indomethacin for three weeks to evaluate lipid accumulation in cells. The cells were fixed to 70% ethanol at room temperature for 15 min and then treated with 2% Oil RedO reagent at room temperature for 1 h. To visualize lipid droplets, the cells were washed with 70% ethanol, and the excess stains were removed through washing with distilled water. In addition, TMSCs were cultured in a medium treated with osteogenesis-induced reagents (10% FBS, 0.1 mM dexamethasone, 10 μM glycerophosphate, and α-MEM supplemented with 50 μg/mL ascorbic acid) to induce osteocyte differentiation. After fixing the cells in 70% ethanol and washing with distilled water, the cells were incubated at room temperature for 15 min in 2% Alizarin Red S solution and then washed several times with distilled water.

### 2.8. Flow Cytometry

We used flow cytometric analysis to measure the overall and active cell surface ITG11 levels of TMSCs. The cells (5 × 10^4^) were incubated with phycoerythrin-conjugated monoclonal antibodies against human guns and active ITG11 in 100 μL of PBS containing 0.5% bovine serum albumin and 2 mmol/LEDTA. The labeled cells were analyzed by cell analysis using a FACS Caliber leucocyte analyzer equipped with CellQuest Pro software (BD Biosciences).

### 2.9. Immunocytochemical Staining

The cells were fixed in 4% paraformaldehyde for 10 min, followed by incubation in 0.5% Tween 20 for 10 min to ensure permeability. The cells were incubated in a solution containing 1% bovine serum albumin for 1 h at room temperature, and then stained with purified rabbit anti-human ITG β11 at 4 °C for 2 h, followed by incubation with phycoerythrin-conjugated chlorine anti-rabbit IgG (e-Bioscience, San Diego, CA, USA) and FITC-conjugated anti-mouse IgG (e-Bioscience) for 1 h at 21 °C. The cells were stained with 4′,6-diamidino-2-phenylindole in VECTASHIELD (Vector Laboratories, Burlingame, CA, USA) mounting medium and examined using a confocal laser scanning microscope.

### 2.10. Statistical Analysis

The data are shown as the mean ± standard deviations for all experiments. One-way analysis of variance (SPSS version 18.0 software, SPSS, Inc., Chicago, IL, USA) and Scheffé’s tests were used to detect significant differences between the groups. A *p*-value < 0.05 was considered to indicate statistically significant results.

## 3. Results

### 3.1. Tensin1 and 3 Control the Proliferation and Migration of TMSCs

Differences in the cell growth after the suppression of various proteins in the TNS family were compared to investigate the effect of TNS on the proliferation of TMSCs (Figure 1). RT-PCR analysis revealed the significantly lowered TNS expression upon transfection with siRNA TNS (siTNS), confirming the successful silencing of TNS mRNA. TNS inhibition only resulted in a difference in the cell proliferation rate, and cell shapes remained unaltered, as confirmed by phase-contrast micrography. Suppression of *TNS1* expression significantly decreased the cell count from day 2 (Figure 1A). siTNS2 transfection-induced *TNS2* inhibition decreased the cell proliferation rate. However, the difference was less than that of *TNS1* inhibition (Figure 1B). The cell proliferation was significantly different before and after *TNS3* inhibition. When *TNS3* was inhibited, the number of cells significantly decreased on days 2 and 3 (Figure 1C). In contrast, a marginal difference was observed in the number of cells before and after transfection with Si *TNS4* (Figure 1D). Table 1 shows the doubling times after TNS family. These results suggest that *TNS3* and *TNS1*, and not *TNS4*, are involved in regulating TMSC proliferation. To understand how TNS1 and TNS3 affect proliferation, we investigated the changes in p16, p19, p21, and cyclin-dependent kinase (CDK) inhibitors associated with cell-cycle arrest and cellular senescence after TNS inhibition (Figure 2). P16 and p21 arrest the cell cycle by interfering with cyclin and CDK complexes in G1 and S phases. P19 interferes with the cyclin D-CDK4/6 complex, delaying cell cycle progression in the G0/G1 phase. TNS1 knockdown increased the p16, p19, and p21 gene and protein expression levels.

Notably, p21 expression was significantly increased at the gene and protein levels. Cyclin E and CDC25 expression also increased (Figure 2A). *TNS3* inhibition resulted in changes similar to those observed for *TNS1* inhibition (Figure 2C). In contrast, *TNS2* inhibition did not result in significant changes in the expression levels of p16, p19, and p21 (Figure 2B). *TNS4* inhibition slightly decreased the expression levels of p16, p19, and p21, although the changes in the gene or protein expression were not significantly different. In contrast, RT-PCR analysis revealed an increased expression of CDC25 and cyclin E (Figure 2D). These findings collectively suggest that TNS1 and TNS3 promote proliferation by accelerating the cell cycle by controlling the CDK inhibitor, a cell-cycle arrest marker.

Inhibiting the four members of the TNS family tended to decrease the cell migration rate (Figure 3). Notably, the silencing of *TNS1* and *TNS3* significantly reduced cell migration. This suggests that TNS1 and TNS3 have significant effects on cell migration.

### 3.2. TNS Also Influences the Pluripotency of TMSCs

We investigated the effect of TNS on the pluripotency of TMSCs using the transcription factors *Nanog*, Sox-2, and Oct4, which maintain and regulate stem cell pluripotency (Figure 4). When TNS1 was silenced, Oct4 and Sox-2 levels showed a slight but non-significant decrease. Conversely, Nanog expression was significantly increased (Figure 4A). When *TNS2* was inhibited, Oct4 and Nanog levels were significantly decreased, but Sox-2 levels were not significantly different (Figure 4B). Transfection with siTNS3 significantly reduced the expression levels of Oct4, Nanog, and Sox-2 (Figure 4C). Conversely, when TNS4 was silenced, Oct4 and Sox-2 significantly increased, whereas the Nanog did not change (Figure 4D). Collectively, these results suggest that the four TNSs affect the pluripotency of TMSCs through appropriate role sharing. Additionally, TNS4 has negative control over Oct4 and Sox-2.

### 3.3. Tensin Controls Osteogenic and Adipogenic Differentiation of TMSC

TMSCs have superior osteogenic and adipogenic differentiation potential as compared to those of other MSCs. To investigate the effect of TNS on the differentiation of TMSC, osteogenic and adipogenic differentiation were investigated after inhibiting TNS (Figure 5). First, TMSCs were cultured in an osteogenic differentiation medium and then observed on days 5, 10, 15, and 20 after Alizarin Red S staining to estimate calcium deposits. In siControl-treated TMSCs, the staining was visible on day 10, and the intensity gradually increased until day 20. In the siTNS1-treated cells, weak staining with minimal calcium deposits was observed on day 20. RT-qPCR analysis revealed that osteogenic differentiation markers, osteocalcin and ALP, decreased significantly from day 10 in siTNS1-treated cells compared to that in siControl-treated cells. When TMSCs were cultured in an adipogenic differentiation medium and stained with Oil Red O, typical adipocytes and lipid droplets were observed in the SiControl-treated cells from day 10. In contrast, lipid droplets were significantly reduced in SiTNS1-treated cells during the entire study period. In RT-qPCR analysis, key transcription factors related to adipogenesis, PPARγ, and LPL, were downregulated in siTNS1-treated cells compared to that in control cells after 15 and 20 days of adipogenic treatment (Figure 5A). SiTNS2-transfected TMSCs showed no differentiation from siControl TMSCs in both osteogenesis and adipogenesis. When cultured in osteogenic and adipogenic differentiation media, siTNS2-transfected cells demonstrated no changes in calcium deposits and lipid droplets compared to those in siControl cells. RT-PCR analysis revealed no changes in the markers of osteogenesis and adipogenesis (Figure 5B). The inhibition of TNS1 and TNS3 yielded similar results in terms of osteogenesis and adipogenesis. Osteogenic and adipogenic differentiation decreased in the siTNS3 group compared to those in the siControl group (Figure 5C). Transfection with siTNS4 demonstrated no change in osteogenic differentiation but unexpectedly led to an increase in adipogenic differentiation. When siTNS4-treated TMSCs were cultured in osteogenic differentiation media, no difference was observed in Alizarin Red S staining and osteocalcin and ALP expression compared to that in siControl TMSCs. However, siTNS4-treated TMSCs cultured in adipogenic differentiation media had an increased number of lipid droplets compared to that in siControl TMSCs. PPARγ and LPL expression significantly increased from day 10 (Figure 5D). Changes in Wnt signaling were investigated to elucidate the pathway related to the inhibition of adipogenic differentiation by TNS4 (Appendix A). The expression or nuclear translocation of β-catenin in siTNS4-treated cells remained unaltered. When *TNS4* was inhibited, pERK was downregulated, whereas pAkt, an essential factor in adipogenesis, did not change (Appendix A).

Collectively, TNS1 and TNS3 play a role in promoting osteogenic and adipogenic differentiation, whereas TNS4 is involved in inhibiting adipogenic differentiation of TMSCs by regulating Akt and ERK signaling.

### 3.4. TNS3 Regulates the Activity and Distribution of Integrin β1

To investigate the effect of TNS on focal adhesions, changes in integrin activity were measured after TNS inhibition (Figure 6). The total and active ITGβ1 levels on the cell surface of siTNS1-treated and siControl-treated TMSCs were compared using flow cytometry. The total ITGβ1 levels were 81.73% ± 1.06% in the siControl-treated TMSCs and significantly reduced to 69.25 ± 2.48% following siTNS1 treatment. Active ITGβ1 levels did not change significantly after siTNS1 treatment (79.83 ± 2.59% to 72.77 ± 1830%). After treatment with siTNS1, red-stained ITGβ1 showed decreased immunofluorescence staining, but there was no difference in its distribution (Figure 6A). Although *TNS2* and *4* were inhibited, the total and active ITGβ1 remained unaltered, as revealed by flow cytometry, and no differences were observed in immunofluorescence staining (Figure 6B,D). Transfection with siTNS3 led to a significant reduction in both total and active ITGβ1 in flow cytometry (74.72 ± 12.56 to 43.97 ± 6.35 and 72.54 ± 15.08 to 42.68 ± 7.03, respectively). The inhibition of *TNS3* also weakened active ITGβ1 staining by immunofluorescence staining. Notably, ITG, which was mainly distributed at the edge of the cell, was expressed near the nucleus in the siTNS3 cells (Figure 6C). This suggests that TNS3 is involved in controlling focal and fibrillar adhesions by regulating integrin activity and distribution.

The expression of talin1 and kindlin2, well-known integrin regulators, significantly decreased when *TNS1* and *3* were inhibited. However, when *TNS4* was inhibited, a slight but non-significant increase was observed (Figure 7). Among the signaling pathways associated with focal adhesion, pERK, pATK, pFAK, and pJNK decreased in siTNS1-, 2-, and 3-treated cells. In contrast, in siTNS4-transfected cells, pATK and pFAK remained unaltered, and pERK and pJNK decreased (Appendix A).

## 4. Discussion

The role of focal adhesion molecules, such as talin and kindlin, and their relationship with integrins in stem cells have been actively studied in recent years [2,9,15]. However, relatively little is known about the role of TNS in focal adhesions. Here, we demonstrated that the TNS family regulates biological processes, including proliferation and differentiation of TMSCs, through integrins at focal adhesions. The two most critical factors for the clinical value of stem cells are self-renewal and pluripotency. We previously showed that TMSCs are significantly superior in proliferation and differentiation potential compared to conventional AMSCs or BMSCs [11,13]. In addition, we demonstrated significantly higher TNS3 in TMSCs than in AMSCs and BMSCs, and that TNS3 affects the proliferation and differentiation of TMSCs [13].

This study investigated the effect of the entire TNS family, including TNS1, 2, 3, and 4, on TMSC biology. TNS1, 2, and 3 showed similar results, but TNS4 showed varying results. When *TNS1*, *2*, and *3* were inhibited, TMSC proliferation was significantly reduced. However, *TNS4* did not have a discernable effect on proliferation. Similarly, no significant change was observed in the cell cycle or pluripotency with TNS4 inhibition. Unlike TNS1 and 3, which affected ITGβ1 activity, TNS4 did not alter ITGβ1 activity. The suppression of *TNS1* and *3* decreased osteogenic and adipogenic differentiation, whereas the inhibition of *TNS4* increased adipogenic differentiation. We investigated the changes in Wnt signaling to reveal the pathway by which TNS4 is involved in adipogenesis. However, we did not observe any change in the expression and nuclear translation of β-catenin. In siTNS4, Akt phosphorylation remained unaltered, whereas ERK phosphorylation and AMPK were decreased, which are thought to increase adipogenic differentiation.

TNS plays a role in transferring structural and chemical changes in cells through integrin to the inside and outside of cells during focal adhesions. To perform this role, it has a multi-domain protein that can bind to various signaling molecules present in focal adhesion, as well as the cytoplasmic tail of ITGβ1 [14]. TNS is structurally divided into three regions. The N-terminus contains an actin-binding domain (ABD) that interacts with actin filaments. The opposite C-terminus contains the Src homology 2 (SH2) domain that can bind to tyrosine-phosphorylated proteins such as FAK, PI3 kinase, MET, and p130Cas [16]. C-terminus also has a phosphotyrosine-binding (PTB) domain, where it binds to the cytoplasmic tail of ITGβ1 [17]. In addition, focal adhesion-binding (FAB) sites are present in both the N- and C-terminal regions. Except for the protein kinase C (C1) domain of TNS2, the structures of the C-terminus and N-terminus of TNS1, 2, and 3 are almost identical [18,19]. However, TNS4, also called a COOH-terminus TNS-like molecule (CTEN), does not have an N-terminal region, unlike the other three TNSs. Therefore, TNS4 does not bind to actin due to the absence of ABD, and it plays a functionally different role from that of TNS1, 2, and 3 because of its conformational specificity [20,21]. TNS1, 2, and 3 play a role in the maturation of focal adhesions and progression to fibrillar adhesions by stably linking actin and integrin [16,22,23]. On the other hand, TNS4 are known to promote cell migration by uncoupling between integrin and actin cytoskeleton. TNS4 may also have a different relationship with the tyrosine-phosphorylated proteins, which interact through the SH2 domain, unlike other TNS families. TNS4 directly interacts with MET through the SH2 domain and regulates MET-dependent cell migration. The downregulation of TNS4 reduces the level of cell-surface expression of MET and affects MET downstream signaling, but the silencing of TNS3 did not significantly affect MET [20].

TNS3 had the most significant influence on most biological processes, including the proliferation and differentiation of TMSCs. TNS3 treatment also led to the most substantial change in ITGβ1 activity. TNS1 exhibited results similar to those of TNS3 in almost all experiments. However, the range of the change was smaller than that of TNS3. TNS2 did not affect migration or differentiation, except for TMSC proliferation. The N- and C-terminal regions of TNS1, 2, and 3 were almost identical, with differences observed in the middle region [14]. Further investigation is required to ascertain whether the different roles of TNS1, 2, and 3 are due to the divergence in this middle region or the high expression rate of TNS3 in TMSCs.

Various biological functions of TMSCs are likely achieved through the cooperation among TNS families with a similar structure but play slightly different roles in focal adhesion. For example, TNS3 plays a vital role in focal adhesion, and a stable integrin–TNS–actin complex is formed through the ABD and PTB domains on both sides of TNS3. However, due to the downregulation of TNS3 and upregulation of TNS4 caused by external stimuli, such as the epidermal growth factor, TNS3 is replaced with TNS4 without ABD. Consequently, the complex with actin stress fiber, which maintains the cell shape, is disrupted, and cells begin to migrate. This “reciprocal TNS3-TNS4 switch” is thought to play a critical role in various biological processes, including cell migration at focal adhesions [21,24].

This study had several limitations. First, only the effects of TNS on integrin and focal adhesion kinases were investigated, and the reciprocal regulation loop between TNS and tyrosine-phosphorylated proteins was not investigated. There we could not determine how TNS4 exhibits different actions from other TNSs. Furthermore, to examine the effect of TNS on the biological process of TMSCs, only the inhibition by siRNA was investigated, and overexpression-mediated changes were not estimated. Nonetheless, we comprehensively investigated the effects of four TNSs on the performance of TMSC functions, including proliferation, cell cycle regulation, maintenance of pluripotency, and differentiation for the first time. We further established that TNS3 regulates integrin activity, whereas TNS4 does not. In particular, we revealed that TNS3 and TNS4 have opposing actions during adipogenic and osterogenic differentiation. In order to more precisely control the differentiation of TMSCs, further studies on the mechanisms of action of TNS3 and TNS4 in focal adhesion during differentiation will be conducted, and we expect that this study will serve as a foundation for future therapeutic applications of TMSC.

## 5. Conclusions

The TNS family plays a vital role in various biological processes in MSCs, including proliferation and differentiation. Although TNS1 and 3 generally perform similar functions, TNS3 in particular has the most potent effect by regulating integrin activity at focal adhesions resulting in the superior proliferation and differentiation potential of TMSCs compared to that in other MSCs. TNS4 plays a distinct role from the other TNSs. TNS4 is not involved in proliferation and inhibits TMSC adipogenesis.

## Figures and Tables

**Figure 1 cells-11-02333-f001:**
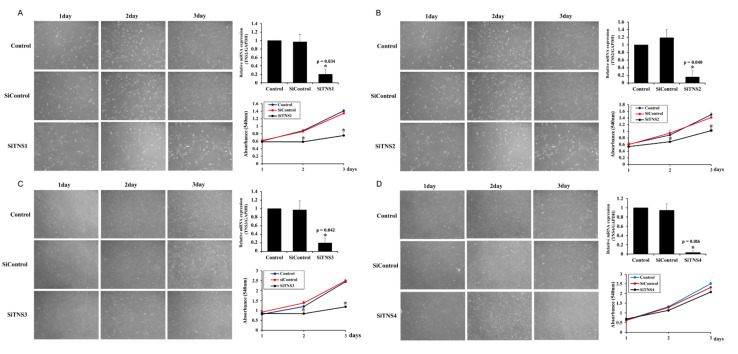
Tensin family regulates the proliferation of tonsil mesenchymal stem cells. Photomicrographs after siRNA TNS1 (siTNS1) (**A**), siTNS2 (**B**), and siTNS3 (**C**) transfection showed a significant reduction in cell numbers on days 2 and 3. However, no morphological changes were observed in transfected cells compared to those in the control. (**D**) Photomicrographs after TNS4 suppression showed no difference in cell number compared to that in the control. * *p* < 0.05, compared to the control. *n* = 4. *Columns* and *error bars* represent mean ± standard deviation.

**Figure 2 cells-11-02333-f002:**
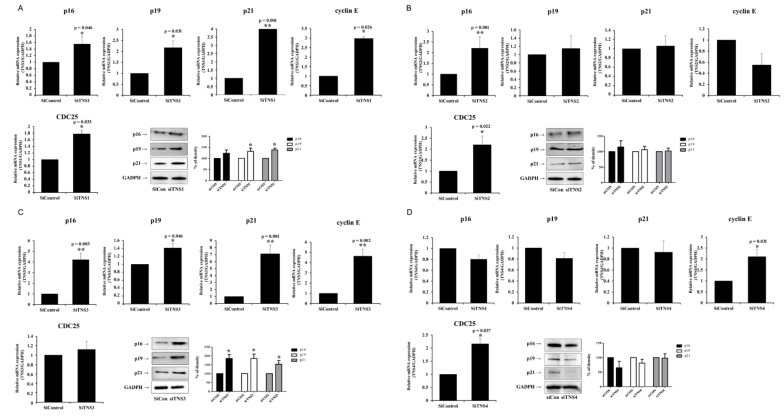
Tensin1 and 3 control cell-cycle regulators in tonsil mesenchymal stem cells. After siTNS1 (**A**) and siTNS3 (**C**) treatment, RT-PCR and Western blotting analysis confirmed the increased expressions of p16, p19, and p21. (**B**) No significant changes were observed in markers for cell-cycle arrest in siTNS2-treated cells. (**D**) After siTNS4 transfection, p16, p19, and p21 decreased slightly, but there was no significant difference. * *p* < 0.05 and ** *p* < 0.01, compared to siControl. *n* = 4. *Columns* and *error bars* represent mean ± standard deviation.

**Figure 3 cells-11-02333-f003:**
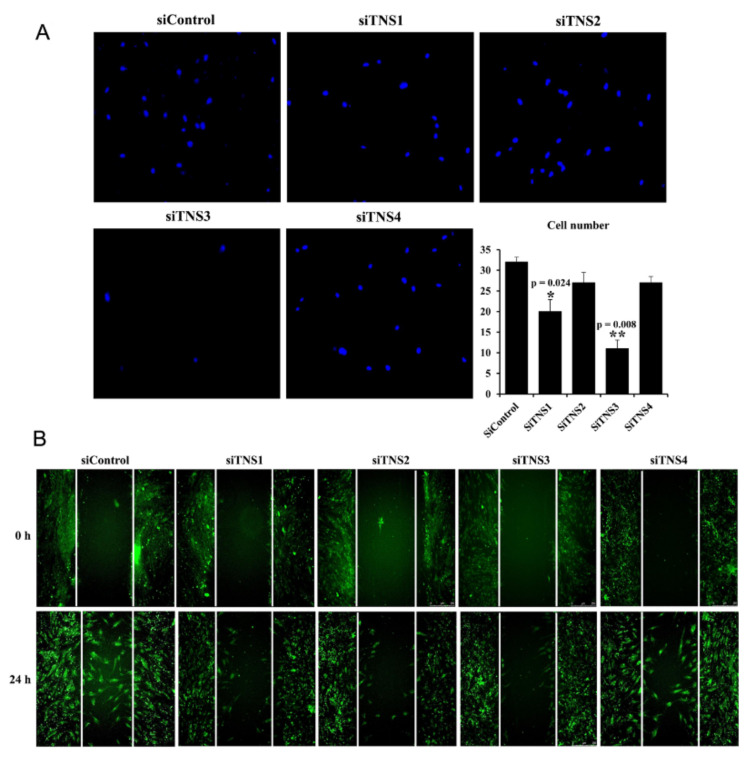
Tensin family is involved in the migration of tonsil mesenchymal stem cells. Migration cell number was reduced after transfection with siTNSs in both transwell (**A**) and two-dimensional wound healing assay with fluorescence on actin (**B**). Notably, it was significantly decreased after siTNS1 and siTNS3 treatment. * *p* < 0.05 and ** *p* < 0.01, compared to siControl. *n* = 4. *Columns* and *error bars* represent mean ± standard deviation.

**Figure 4 cells-11-02333-f004:**
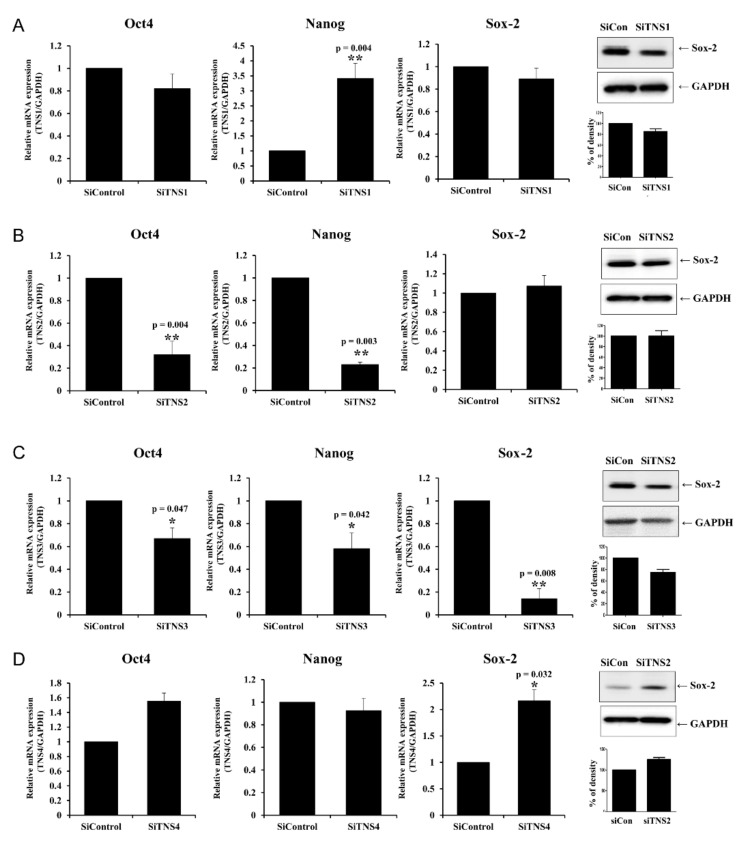
Tensin also influenced the pluripotency of tonsil mesenchymal stem cells. The graph depicts the changes in Oct4, Nanog, and Sox-2 after treatment with siTNS1 (**A**), siTNS2 (**B**), siTNS3 (**C**), and siTNS4 (**D**). Oct4 and Nanog were significantly decreased upon siTNS2 and siTNS3 treatment. siTNS3 and SiTNS4 treatment decreased and increased Sox-2 levels, respectively. * *p* < 0.05 and ** *p* < 0.01, compared to siControl. *n* = 4. *Columns* and *error bars* represent mean ± standard deviation.

**Figure 5 cells-11-02333-f005:**
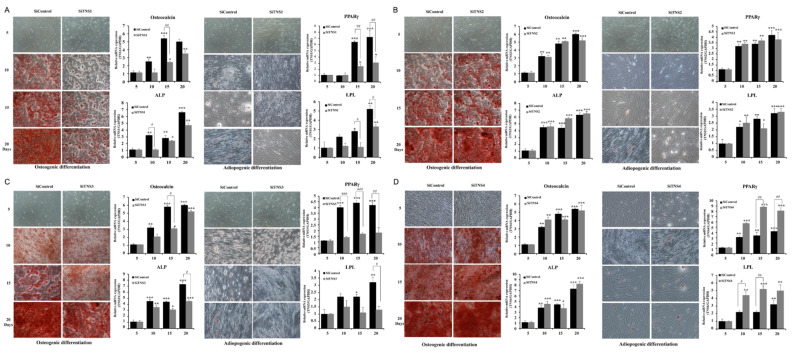
Tensin controls osteogenic and adipogenic differentiation of tonsil mesenchymal stem cells (TMSCs). (**A**) In the osteogenic differentiation medium, siTNS1 treatment reduced the Alizarin Red S staining suggesting a lower expression of osteocalcin and ALP from 10 days compared to that in the siControl-treated cells. In the adipogenic differentiation medium, siTNS1-treated cells showed a significantly reduced number of lipid droplets and reduced PPARγ and LPL expression. (**B**) SiTNS2-transfected TMSCs showed no osteogenic or adipogenic differentiation compared to that in siControl TMSCs. (**C**) siTNS1 and siTNS3 demonstrated similar results in terms of osteogenic and adipogenic differentiation. (**D**) Transfection with siTNS4 revealed no change in osteogenic differentiation but led to an increase in adipogenic differentiation. * *p* < 0.05, ** *p* < 0.01, *** *p* < 0.001, compared to five days. ^#^ *p* < 0.05, ^##^ *p* < 0.01, and ^###^ *p* < 0.001, compared to siControl. *n* = 4. *Columns* and *error bars* represent mean ± standard deviation.

**Figure 6 cells-11-02333-f006:**
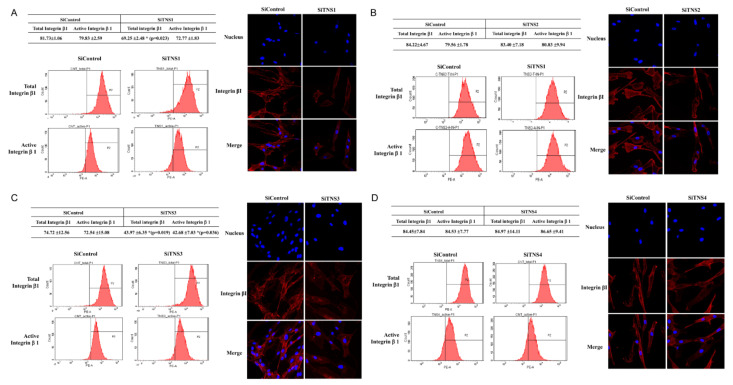
TNS3 in focal adhesion regulates integrin activity in tonsil mesenchymal stem cells. (**A**–**C**) Flow cytometric analysis revealed no significant change in active ITGβ1 levels in siTNS1-, siTNS2-, and siTNS3-treated cells compared to that in siControl-treated cells. (**D**) Transfection with siTNS3 led to a significant reduction in total and active ITGβ1 according to flow cytometry analysis (left panels). siTNS3-transfection also weakened active ITGβ1 staining in immunofluorescence assays. ITG β1, which was mainly distributed at the edge of the cell, was expressed near the nucleus after siTNS3-treatment (right panels). * *p* < 0.05, compared to siControl. *n* = 4.

**Figure 7 cells-11-02333-f007:**
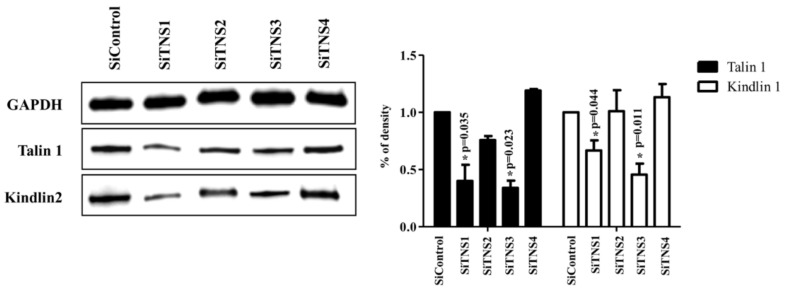
TNS1 and TNS3 interact with talin1 and kindlin2 in focal adhesion of tonsil mesenchymal stem cells (TMSCs). Western blotting analysis suggested that talin1 and kindlin2 were significantly decreased in siTNS1- and siTNS3-treated cells. In contrast, transfection with siTNS4 led to a slight increase in talin1 and kindlin2 levels. * *p* < 0.05, compared to siControl. *n* = 4. *Columns* and *error bars* represent mean ± standard deviation.

**Table 1 cells-11-02333-t001:** Population doubling time after inhibition of tensins.

	SiControl	SiTNS1	SiTNS2	SiTNS3	SiTNS4
**Doubling time (h)**	42.7 ± 3.6	167.0 ± 31.2	109.7 ± 20.8	315.2 ± 54.8	48.8 ± 4.0

## Data Availability

Not applicable.

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
