# Peer review of "Tensin Regulates Fundamental Biological Processes by Interacting with Integrins of Tonsil-Derived Mesenchymal Stem Cells"

_cells, 2022, doi:10.3390/cells11152333_

Round 1
Reviewer 1 Report
In this article the authors conducted a comprehensive study on multiple TNS subdomains with their effect on tonsil-derived mesenchymal stem cells in terms of proliferation, migration, osteogenic/adipogenic differentiation and the relation to focal adhesions. The authors prepared a list of TNS silenced cell lines, reported four major domains and the associated performance for each concept. Among four tested subtypes, TNS1 and TNS3 exhibit similar functions, explicitly the TNS3 is most potent to regulate the integrin activity at focal adhesions. TNS4 is quite different from the other three and in many experiments mentioned it shows a reverse effect which I would personally suggest extensive study. In general this manuscript is near complete for the current stage of research, but it definitely need more input on focal adhesion studies. I would recommend a major revision with extra experimental and discussion to further consolidate a publication quality.
Major questions:
1. The most important question I have is how TNS4 can act so differently from the other three. How TNS4, and of cause the other three, spatiotemporally distributed inside the cell? I would suggest a labeling and staining on all 4 subtypes and track the co-localization with proteins of interest. Also please include more literature study and your thought in the discussion.
2. While the paper intended to focus on the focal adhesion perspective, I am looking for some other biomarkers such as paxillin, FAK, Arp2/3 and even actin bindings. The progression of actin network anchored to the focal adhesion would be of great value for migration, proliferation and differentiations. The turnover of focal adhesion should also be affected by TNS which could help identify any potential pathways it belongs to.
3. In what stage during focal adhesion maturation does TNS play the role? This might be related to the 2nd item above, but I would like to see more insight on the dynamics/kinetics from a live cell image on the basal layer, not in the middle where you label nucleus as well.
4. Consider doing a 2D migration assay with fluorescence on either cytoskeletal structure (e.g. actin) or membrane, show the trajectory and high resolution images to better illustrate the effect of TNS on migration/focal adhesion.
Minor questions:
1. I suggest put scatter on top of boxplot whenever possible, and clearly label the p-values.
2. Lots of typos on ‘GAPDH’ on the western blots. Also we need different GAPDH bands for each gene and protein expression when they are not performed at the same time. This could be shown in supplemental but is mandatory.
3. Enlarge your plots, I cannot really see it clearly.
4. Extra quantifications on cell proliferation other than abundance plot, and the doubling time.
Author Response
We really appreciate the constructive and helpful comments of reviewers. We have carefully addressed the comments and have revised the manuscript as suggested. Our responses are given point-by-point manner below. We made every effort to improve our manuscript based on the suggested comments.
Our point-by-point responses are attached as a pdf file.

Reviewer 2 Report
In this paper, the role of different isoforms of TNS in different biological processes of TMSCs and their relationship with integrins were studied, and the effects of different TNS on the migration and differentiation of TMSCs were discussed. The results have a good reference value for controlling the directional differentiation of TMSCs at the gene level. In view of this, it is suggested that the article can be published in this journal after the following modifications:
1. The shape of the cells in figure 1 (D) is not clear, so it is difficult to judge the number and state of the cells. The contrast of the picture should be adjusted to facilitate mutual verification with the narrative results.
2. On the fifth page of the article, before explaining the effects of TNS on P16, P19 and P21, the role of these genes in explaining cell proliferation should be briefly introduced, so as to facilitate the readers to understand the significance of the work.
3. What is the significance of studying the role of different isoforms of TNS in controlling cell differentiation in the future? It is suggested that the author should make a prospective statement on this.
Author Response

(The authors gave the same response as above.)

Round 2
Reviewer 1 Report
Most of the items I wrote are properly responded with either revision on manuscript or new experiments. I support the publication of this form, and hope to see their next stage of work.